# Synergistic Integration of Skeletal Kinematic Features for Vision-Based Fall Detection

**DOI:** 10.3390/s23146283

**Published:** 2023-07-10

**Authors:** Anitha Rani Inturi, Vazhora Malayil Manikandan, Mahamkali Naveen Kumar, Shuihua Wang, Yudong Zhang

**Affiliations:** 1Department of Computer Science and Engineering, SRM University—AP, Mangalagiri 522240, India; anitha_rani@srmap.edu.in (A.R.I.); naveenkumar.m@srmap.edu.in (M.N.K.); 2School of Computing and Mathematical Sciences, University of Leicester, Leicester LE1 7RH, UK; sw546@le.ac.uk

**Keywords:** fall detection, video analysis, vision-based human activity recognition, fall prevention, ambient intelligence, assistive technology, signal processing, real-time monitoring, risk assessment

## Abstract

According to the World Health Organisation, falling is a major health problem with potentially fatal implications. Each year, thousands of people die as a result of falls, with seniors making up 80% of these fatalities. The automatic detection of falls may reduce the severity of the consequences. Our study focuses on developing a vision-based fall detection system. Our work proposes a new feature descriptor that results in a new fall detection framework. The body geometry of the subject is analyzed and patterns that help to distinguish falls from non-fall activities are identified in our proposed method. An AlphaPose network is employed to identify 17 keypoints on the human skeleton. Thirteen keypoints are used in our study, and we compute two additional keypoints. These 15 keypoints are divided into five segments, each of which consists of a group of three non-collinear points. These five segments represent the left hand, right hand, left leg, right leg and craniocaudal section. A novel feature descriptor is generated by extracting the distances from the segmented parts, angles within the segmented parts and the angle of inclination for every segmented part. As a result, we may extract three features from each segment, giving us 15 features per frame that preserve spatial information. To capture temporal dynamics, the extracted spatial features are arranged in the temporal sequence. As a result, the feature descriptor in the proposed approach preserves the spatio-temporal dynamics. Thus, a feature descriptor of size [m×15] is formed where *m* is the number of frames. To recognize fall patterns, machine learning approaches such as decision trees, random forests, and gradient boost are applied to the feature descriptor. Our system was evaluated on the UPfall dataset, which is a benchmark dataset. It has shown very good performance compared to the state-of-the-art approaches.

## 1. Introduction

Machine learning algorithms, in combination with video processing, analyze the videos and identify the human activity. This may entail activities like scene segmentation, object detection, object tracking, human activity recognition and others. The information contained in the video is automatically extracted with the power of machine learning algorithms. The extracted video data are used for various applications such as surveillance, medicine, criminal activity recognition, assisted living, military and many more. Vision-based fall detection is one such application of video processing that alerts people in case of a fall and has a significant demand in assisted living environments. Falls are a serious risk, especially for adults over 60 owing to their cognitive decline and cell degeneration [1]. Age-related physiological, psychological, neurological and biological changes in elders are the inherent factors for falling. In general, falls occur due to multiple reasons such as dizziness, chronic illness, vision impairment, gait or other environmental conditions such as slippery floors, highland terrain, outfit etc.

According to centers for disease control and prevention, falling is a major cause of fatality, and the fatality rate is increasing rapidly worldwide, especially among senior citizens. Every year, there are 36 million recorded falls, 3 million of which require emergency room treatment, 95% of which result in hip fractures, and 32,000 deaths are reported [2]. The consequences of falls, which may be social, physical, or psychological, are depicted in Figure 1.

This leads to an increased focus on researching fall detection, highlighting the significance of identifying falls. Consequently, the research community predominantly uses multivariate data in their efforts to develop fall detection methods.

Falls can be detected using both sensor-based [3,4,5] and computer vision-based [6,7,8] technologies. Sensor-based technologies rely on input from wearable sensors worn by the subject. Computer vision-based technologies analyze the data obtained from a monitoring camera. While each method has particular drawbacks, they are both technically capable of identifying falls. Figure 2 depicts the general workflow of a fall detection system. In the model distribution phase, A1. A2, A3, A4 represent the four quadrants where data may be distributed. The feature vector is also a collection of different features from frame1 to framen.

Input sources for sensor-based systems typically include accelerometers, magnetometers, and gyroscopes, which are usually embedded in smartwatches, smartphones, and other wearable gadgets. These systems extract features such as speed, velocity, field orientation, variation of the magnetic field, and angular momentum. Contrarily, computer vision-based systems often employ Kinect or depth cameras to record input data in the form of RGB or depth images. The characteristics that can be utilized to detect falls are extracted by these systems after they have analyzed the images or frames. These features are as follows:iLocal features such as the colour, texture, and intensity of the imageiiGlobal features such as the image silhouette, edges, spatial pointsiiiDepth features, that extract the depth information of the image.

Our main contribution lies in the design of a new feature vector that results in a new fall detection framework. In the proposed approach, we defined a set of new informative features through the following steps:Segmenting the human skeleton into five sections (left hand, right hand, left leg, right leg, and a craniocaudal section).Extracting the distances from the segmented parts (Spatial domain).Calculating the angles within the segments (Spatial domain).Calculating the angle of inclination for every segment (Spatial domain).To capture temporal dynamics, the extracted spatial features are arranged in the temporal sequence. As a result, the feature descriptor in the proposed approach preserves the spatiotemporal dynamics.We have achieved very good performance compared to the state-of-the-art approaches.The performance of our method is evaluated on the UPfall dataset.

In this paper, we proposed a new fall detection system. A comprehensive description of the entire process is given below.

By extracting *m* frames from the UPfall dataset using Equation (Equation 1).The AlphaPose pre-trained network was applied to retrieve 17 keypoints of the subject from every frame.The missing keypoints were computed using Equations (Equation 2) and (Equation 3).From the 17 keypoints, 13 keypoints were retrieved and two additional keypoints were computed. In total, 15 keypoints were used for the process.These 15 keypoints were used to segment the human skeleton into five sections: left hand, right hand, left leg, right leg, and a craniocaudal section.Three features were extracted from each section. Specifically, the length of the section (distances), the angle made by the points that depict a section and the angle made by every section with the x-axis.As a result, 15 features were retrieved from each frame. Each feature was represented by a column.Thus, we extracted characteristics from *m* frames and aligned them column-wise so that each row represents one video frame in a temporal sequence.Hence a feature descriptor was formed. This descriptor was the input to the machine learning algorithms.Also, to preserve the ground truth, every video was labelled as a fall or not a fall.The accuracy of the machine learning algorithms was computed using the ground truth data.

The structure of the paper is as follows: Related work is discussed in Section 2, the proposed methodology is presented in Section 3, experimental results are outlined in Section 4, and finally, Section 5 concludes the paper and proposes areas for future research.

## 2. Literature Review

In this section, we are discussing a few existing fall detection approaches. In the field of computer science, computer vision [9,10,11] and machine learning [12,13] are widely used for solving various problems, such as sentiment analysis [14], speech recognition [15,16], and image processing [17,18,19]. Combining computer vision, machine learning, and deep learning has proven to be very effective in resolving a multitude of problems, including human activity recognition (HAR). HAR involves recognizing a person’s actions, such as jumping, running, laying, bending, walking, and sitting [20,21,22]. Detecting falls is a crucial aspect of activity recognition, but it can be a difficult task due to the subtle differences between falls and other activities such as bending or lying down. There are several approaches to fall detection, and we will discuss some of the key ones below.

### 2.1. Sensor-Based Technology

Fall detection systems process the data that are generated by sensor devices. Sensor devices are more useful in outdoor environments. The authors in their work [23] designed a wearable sensor that detects falls and sends the individual’s aid request and position information to carers using a quaternion algorithm. This system applies the algorithm to acceleration data. Their approach, however, might cause false alarms because they might mistake sleeping or lying down for a fall. A bi-axial gyroscope sensor array-based system with three thresholds applied to changes in trunk angle, angular acceleration, and velocity was suggested by the authors of [24]. In controlled studies on young people, their method produced 100% specificity.

Another work proposed in [25] created a straightforward smart carpet-based system that detects falls using piezoresistive pressure sensors. Their approach acquired a sensitivity and specificity of 88.8% and 94.9%, respectively. A smart floor design that retrieves pressure images for fall detection using a sophisticated back-projection method was proposed in [26]. However, implementing this technology in practical applications is expensive. A new long-short-term memory architecture (LSTM) called cerebral LSTM was introduced by [27] on wearable devices to detect falls. A millimetre wave signal-based real-time fall detection system called mmFall was proposed by [28]. It achieved high accuracy and low computational complexity by extracting signal fluctuation related to human activity using spatial-temporal processing and developing a light convolutional neural network.

### 2.2. Vision-Based Technology

Computer vision systems analyze video feeds from cameras placed in a room to track the subject’s movements to identify falls. Every time there is a change in the subject’s movement, machine learning or computer vision algorithms are used to analyze the patterns and detect a fall. To detect falls, multiple cameras are utilized to record various types of images.

RGB Images: Red, green, and blue (RGB) images include three color channels and can be used to detect falls by examining alterations in the subject’s position and orientation inside the image. A fall can be categorized as a divergence from usual patterns by tracking the movement of keypoints or additional factors.

In [29], the importance of simultaneously tracking the head and torso regions for fall detection is discussed. Geometrical features are extracted from elliptical contours applied to these regions, and a CNN is used to analyze the correlation of these features and detect falls. However, since falls can be similar to a lying position, tracking only the head and torso may result in false positives.

To preserve the privacy of the subject, Reference [30] uses background subtraction to retrieve the human silhouette, which is then stacked and analyzed for binary motion. Similarly, in [31], human silhouettes are extracted instead of raw images. A pixel-wise multi-scaling skip connection network is used to extract the silhouette, which is then analyzed using convLSTM for fall detection. The authors report an excellent f1-score of 97.68%. However, the human silhouettes may include shadows, which could potentially lead to misclassification of falls.

The work proposed in [32] extracted the skeleton keypoints using the AlphaPose network and applied random forest, support vector machines, multi-layer perceptron, and k-nearest neighbours algorithms. They achieved an accuracy of 97.5%. The authors in [33] proposed a fall detection method by tracking the keypoints in the successive frames. In their approach, they computed the distance and angle between the same keypoints in successive frames. Their system was evaluated on the URfall dataset and they achieved an accuracy of 97%. An overview of existing approaches, the methods adopted, and the machine learning algorithms used are given in Table 1.

Depth Images: Depth images, which measure the distance between objects in the frame and allow for the monitoring of changes in an object’s depth, are used to detect falls. In [34], skeletal information is tracked in depth images to enhance privacy preservation. A fall is presumed to have occurred if the head’s motion history images show greater variance in head position over time. But relying only on the head position can result in false positives, for instance when the subject is simply lying down.

A fall detection method that utilizes both RGB and depth images has been proposed in [35]. The RGB images are used to identify feature points as either static or dynamic, while the depth images are clustered using k-means. After that, the RGB features are projected onto the depth images and categorized as static or dynamic. This classification is then used for object tracking. This object tracking can then be utilized for fall detection.

## 3. Proposed Approach

Fall detection systems must respond quickly to prevent significant consequences. Our goal is to improve system performance and minimize response time. To achieve this, we propose a fall detection architecture that reduces computational complexity. We found that computer vision-based approaches are more accurate than sensor-based approaches, as it is not feasible for individuals to wear sensing devices at all times. Therefore, we adopted a computer vision-based approach and proposed features that can accurately distinguish falling from non-fall activities by analyzing the 2D representation of the human skeleton. The use of 2D representation also reduces computational complexity, improving system efficiency.

### 3.1. Dataset

The performance of our fall detection system was evaluated using the UP Fall dataset [36]. This dataset is a collection of multimodal data. It contains data obtained from sensors as well as data obtained from two cameras. The setup consists of frontal and lateral cameras. A total of 17 young and healthy subjects performed eleven activities that are a combination of five types of falls and six daily living activities. Three trials were conducted on every subject.

### 3.2. Pre-Processing

The dataset has been fine-tuned to collect *m* frames from every video using the following Equation (Equation 1)
(1)$$skip\_value=\frac{n}{m}$$
where *n* = total number of frames; *m* = number of frames required.

The regional multi-person pose estimation network (RMPE) [37] or the pre-trained AlphaPose network, which extracts the keypoints of the human skeleton, is utilized to process the fine-tuned dataset. These keypoints indicate the locations of the joints in the human body. The COCO dataset [38] was used to train the AlphaPose network. Figure 3 illustrates the architecture of AlphaPose, which consists of three components.

iThe SSTN + SPPE method, which combines a symmetric spatial transformer network (SSTN) with a parallel single-person pose estimation (SPPE) method, is used to create pose recommendations from human bounding boxes. SPPE is used in parallel with SSTN to control the output when it is unable to provide the desired pose.iiA technique known as parametric pose non-max-suppression (NMS) is used to find similar poses in the dataset and choose the one with the highest score to prevent redundant poses. The dataset is streamlined in this way, and only the poses that are most accurate and pertinent are kept.iiiThe system’s accuracy and robustness are increased using a technique called the pose-guided proposals generator. It operates by locating the human object in the scene and suggesting several bounding boxes that correspond to the various stances the person might strike. These bounding boxes are created using computer vision techniques that estimate the positions of the human joints. This method enables the system to record a large range of potential poses and motions.

We used the AlphaPose architecture to locate spatial coordinates of human joint positions on the human skeleton in the video data from the UP-FALL detection dataset. To determine the locations of the joints in the human body, this architecture uses a model that generates 17 key points. As they represent the critical body positions that can indicate a fall, these key points serve as crucial information for fall detection. We can examine the human body movements and assess if those movements are related to falls or not. However, due to the position of the subject in certain frames, some key points were found to be missing. To address this issue, we employed an imputation method that calculates the average to estimate the missing key points, as shown in Equations (Equation 2) and (Equation 3).
(2)$$F_{i}K_{n}\left(x\right)=\frac{F_{i-1}K_{n}\left(x\right)+F_{i+1}K_{n}\left(x\right)}{2}$$
(3)$$F_{i}K_{n}\left(y\right)=\frac{F_{i-1}K_{n}\left(y\right)+F_{i+1}K_{n}\left(y\right)}{2}$$
where, $F_{i}$ is the *i*-th frame and $K_{n}$ is the *n*-th key point. Hence, $F_{i}K_{n}\left(x\right)$ represents the x coordinate of the *n*-th key point in *i*-th frame and $F_{i}K_{n}\left(y\right)$ represents the y coordinate of the *n*-th key point in *i*-th frame. $F_{i-1}$, $F_{i+1}$ are the earlier and later frames of $F_{i}$, respectively.

### 3.3. Assumptions

Generally, human motion can be observed through the movements of the limbs. Therefore, it is reasonable to assume that tracking a person’s limbs can provide additional information about the action being carried out. Hence, we extracted and analyzed the spatial positions of the four limbs. It was also observed that during daily living activities such as walking, sitting, and standing, the craniocaudal axis, which is the segment joining the head and toe, is usually perpendicular to the ground or the x-axis. However, when a person falls, the angle between the craniocaudal axis and the ground (x-axis) is close to zero. To create more precise features using these presumptions, we considered the four limbs and the craniocaudal axis as five individual components and crafted features such as the distance between the points on a component, the angle made by the component with the x-axis, and the angle formed at the center point of each component. From the 17 keypoints obtained through the AlphaPose network, we selected 13 keypoints and 2 other keypoints were computed to generate a 15-keypoint model. The 15-keypoint models shown in Figure 4 were considered for feature extraction.

An approximation of the position of the hip/spine center (SC) is calculated from the spatial positions of LH and RH using Equation (Equation 4).
(4)$$x_{SC}\approx \frac{x_{LH}+x_{RH}}{2},\;\;\;\;\;\;y_{SC}\approx \frac{y_{LH}+y_{RH}}{2}$$

Similarly, the ankle center is calculated from the spatial positions of LA and RA using Equation (Equation 5).
(5)$$x_{AC}\approx \frac{x_{LA}+x_{RA}}{2},\;\;\;\;\;\;y_{AC}\approx \frac{y_{LA}+y_{RA}}{2}$$

The five components used in the analysis are the left hand, left leg, right hand, right leg, and the craniocaudal segment. The left-hand component is defined by the non-collinear points of the left shoulder (LS), left elbow (LE), and left wrist (LW) represented by their Cartesian coordinates. Similarly, the non-collinear points of the right shoulder (RS), right elbow (RE), and right wrist (RW) represent the right-hand component. The left leg component is defined by the non-collinear points of the left hip (LH), left knee (LK), and left ankle (LA), while the right leg component is defined by the non-collinear points of the right ankle (RA), right knee (RK), and right hip (RH). The craniocaudal axis component is represented by the center of the head (HC), hip/spine center (SC), and the midpoint of the ankles (AC). The Equation (Equation 6) entails these five segments.
(6){LS,LE,LW}{RS,RE,RW}{LH,LK,LA}{RH,RK,RA}{HC,SC,AC}

### 3.4. Kinetic Vector Calculation

The final set of features that characterizes the movement patterns of the individual in the video is determined by measuring the distance between the initial and final points in each of the five components, as well as the angle formed between each component and the x-axis. Additionally, the angle formed at the center point of each component is also computed to enhance the precision of the features. The representation of features is shown in Figure 5, Figure 6 and Figure 7.

Distance between the non-collinear points is calculated using the Equations (Equation 7)–(Equation 11).

For left hand
(7)$$D_{Lefthand}=||LS-LW||$$

For right hand
(8)$$D_{Righthand}=||RS-RW||$$

For left leg
(9)$$D_{Leftleg}=||LH-LA||$$

For right leg
(10)$$D_{Rightleg}=||RH-RA||$$

For craniocaudal line
(11)$$D_{craniocaudalline}=||HC-AC||$$

The angle of inclination β between the non-collinear points is calculated using the Equations (Equation 12)–(Equation 16). The reason to use atan while calculating the beta angle is that values in the second quadrant result in negative numbers and values in the third quadrant result in positive numbers when atan function is applied. This variation from positive to negative and again to positive offers a chance to identify some deviation in a regular pattern which is consistently either positive or negative. Identifying this deviation is more crucial for an angle of inclination rather than the angle between the non-collinear points of a segment.

For left hand
(12)$$\beta_{Lefthand}=\tan^{-1}\frac{\left(y_{LS}-y_{LW}\right)}{\left(x_{LS}-x_{LW}\right)}*\frac{180}{pi}$$

For right hand
(13)$$\beta_{Righthand}=\tan^{-1}\frac{\left(y_{RS}-y_{RW}\right)}{\left(x_{RS}-x_{RW}\right)}*\frac{180}{pi}$$

For left leg
(14)$$\beta_{Leftleg}=\tan^{-1}\frac{\left(y_{LH}-y_{LA}\right)}{\left(x_{LH}-x_{LA}\right)}*\frac{180}{pi}$$

For right leg
(15)$$\beta_{Rightleg}=\tan^{-1}\frac{\left(y_{RH}-y_{RA}\right)}{\left(x_{RH}-x_{RA}\right)}*\frac{180}{pi}$$

For craniocaudal line
(16)$$\beta_{craniocaudalline}=\tan^{-1}\frac{\left(y_{HC}-y_{AC}\right)}{\left(x_{HC}-x_{AC}\right)}*\frac{180}{pi}$$

Base angle α between the non-collinear points is calculated using the Equations (Equation 17)–(Equation 21).

For left hand
(17)$$\alpha_{Lefthand}=\left(atan2\left(\left(x_{LS}-x_{LE}\right),\left(y_{LS}-y_{LE}\right)\right)-atan2\left(\left(x_{LE}-x_{LW}\right),\left(y_{LE}-y_{LW}\right)\right)\right)*\frac{180}{pi}$$

For right hand
(18)$$\alpha_{Righthand}=\left(atan2\left(\left(x_{RS}-x_{RE}\right),\left(y_{RS}-y_{RE}\right)\right)-atan2\left(\left(x_{RE}-x_{RW}\right),\left(y_{RE}-y_{RW}\right)\right)\right)*\frac{180}{pi}$$

For left leg
(19)$$\alpha_{Leftleg}=\left(atan2\left(\left(x_{LH}-x_{LK}\right),\left(y_{LH}-y_{LK}\right)\right)-atan2\left(\left(x_{LK}-x_{LA}\right),\left(y_{LK}-y_{LA}\right)\right)\right)*\frac{180}{pi}$$

For right leg
(20)$$\alpha_{Rightleg}=\left(atan2\left(\left(x_{RH}-x_{RK}\right),\left(y_{RH}-y_{RK}\right)\right)-atan2\left(\left(x_{RK}-x_{RA}\right),\left(y_{RK}-y_{RA}\right)\right)\right)*\frac{180}{pi}$$

For craniocaudal line
(21)$$\alpha_{craniocaudalline}=\left(atan2\left(\left(x_{HC}-x_{SC}\right),\left(y_{HC}-y_{SC}\right)\right)-atan2\left(\left(x_{SC}-x_{AC}\right),\left(y_{SC}-y_{AC}\right)\right)\right)*\frac{180}{pi}$$

A feature vector is constructed from the computed features on the video data, which consists of *m* frames per video and 15 features per frame. This results in a total of 15 *m* features per video, as given in Equation (Equation 22).
(22)$$\left\{D_{Lefthand1},\beta_{Lefthand1},\alpha_{Lefthand1}\cdots \cdots \cdots D_{craniocaudalm},\beta_{craniocaudalm},\alpha_{craniocaudalm}\right\}$$

## 4. Experimental Study and Result Analysis

The performance of the fall detection system will be assessed in the following section by comparing the output of various classifiers. Also, a comparison of the performance of the proposed fall detection system using the approaches in the literature is made. The feature set, consisting of 15 features per frame and m=100 frames per video, is generated for the UPfall dataset, which is further utilized for training and testing machine learning models such as gradient boosting, decision tree, and random forest algorithms to classify videos into fall and non-fall categories. The workflow of the system is shown in Figure 8.

### 4.1. Decision Tree Algorithm

One of the classifiers employed for fall detection is the decision tree algorithm. This algorithm is generally used to address both classification and regression problems. The decision tree method generates predictions by using a succession of feature-based splits organized in a tree-like structure. Every node acts as a decision node starting at the root node and leads to a leaf node that represents the ultimate choice. The input space is recursively divided into subsets up until a halting requirement is satisfied, and this is how decision trees are fundamentally constructed.

Entropy, a statistical metric that measures the degree of unpredictability or impurity in a particular dataset node, is used to determine which node should be the decision tree’s root node and is calculated as in Equation (Equation 23)
(23)$$e\left(s\right)=-P_{f}\;logP_{f}-P_{nf}\;logP_{nf}$$
*s* denotes the number of samples; e(s) is the entropy; $P_{f}$ denotes the probability of a fall; $P_{nf}$ denotes the probability of not a fall.

The measure of information gain represents the quantity of information obtained from a specific feature and is utilized to determine both the root and decision nodes in a decision tree algorithm. Information gain (IG) is calculated as in Equation (Equation 24)
(24)$$IG=E_{parent}-E_{children}$$
where $E_{parent}$ is entropy of parent node and $E_{children}$ is the average entropy of child nodes.

### 4.2. Random Forest Algorithm

A random forest is a collection of multiple decision trees, where each tree predicts a class, and the final output of the model is determined by the class that receives the most votes from the individual trees.

### 4.3. Gradient Boost Algorithm

The goal of the gradient boosting algorithm is to improve weak learners’ performance and transform them into strong learners. It also makes use of an ensemble of decision trees, similar to the random forest algorithm. However, it uses a modified version of the AdaBoost algorithm technique. All observations in the AdaBoost algorithm are trained with equal weights. The weights of the hard-to-classify observations are increased with each iteration, while those of the straightforward observations are dropped.

### 4.4. Evaluation Metrics

Performance measurement is crucial for any classification algorithm in machine learning. The performance metrics used to validate our system are as follows:Accuracy: Accuracy is an important measure of the classifier’s performance.It is the ratio that relates to the proportion of precise predictions made compared to all other predictions. Accuracy is defined as given in Equation (Equation 25).
(25)$$Acc=\frac{ncp}{tnp}$$Acc: accuracyncp: number of correct predictionstnp: total number of predictionsConfusionmatrix: The confusion matrix is a statistic used to evaluate the performance of a predictive model that can be provided in either tabular or matrix style. It serves as a visual representation of a classification model’s true positive, false positive, true negative, and false negative predictions, as shown in Figure 9.The number of falls that were reported as falling is known as true positives ($T_{p}$). The amount of non-falls that were anticipated to be non-falls is known as true negatives ($T_{n}$). The term “false positives” ($F_{p}$) refers to the number of non-falls that were mistakenly identified as falling. False negative ($F_{n}$) refers to the number of falls that were assumed not to occur.Precision: Precision is another measure that calculates how accurately the model predicts positive outcomes. From the total number of samples classified as positive, the number of true positive predictions is identified as the model’s precision. A model with high precision has a lower number of false positives. The precision (Pre) is calculated in the Equation (Equation 26).
(26)$$Pre=\frac{T_{p}}{T_{p}+F_{p}}$$Sensitivity: In machine learning, sensitivity refers to the true positive rate. From the total number of positive samples of ground truth, the number of samples predicted as positive defines the model’s sensitivity. A model with high sensitivity has predicted most of the positive samples correctly, resulting in low false negatives. Sensitivity is calculated as in the Equation (Equation 27).
(27)$$Recall=\frac{T_{p}}{T_{p}+F_{n}}$$Specificity: In machine learning, specificity refers to the true negative rate. From the total number of negative samples in the ground truth, the number of samples classified as negative defines the specificity of the model. A model with high specificity has predicted most of the negative instances correctly. The specificity (Spe) is calculated as given in Equation (Equation 28).
(28)$$Spe=\frac{T_{n}}{T_{n}+F_{p}}$$F1-score: The F1-score metric represents the balance between the true positive rate and the precision. It combines the precision and recall scores to get a single score that assesses a predictive model’s overall effectiveness. F1-score ranges from 0 to 1, where 1 indicates the best case. The F1-score is calculated using both precision and recall as given in Equation (Equation 29).
(29)$$F1-score=2*\frac{Pre*recall}{Pre+recall}$$AUC−ROCcurve: The AUC−ROC curve represents the model performance of a binary classifier. AUC refers to the area under the curve and ROC refers to the receiver operating characteristic. The ROC is a probability curve that plots the true positive rate (sensitivity) against the false positive rate (1–specificity) at various classification thresholds. A higher AUC means that the model is better in its classification.

#### Performance of the Algorithms

Performance results of the algorithms used to evaluate the system, i.e., decision tree algorithm, random forest and gradient boost are given in Table 2, the confusion matrix is shown in Figure 10 and the AUC-ROC curve is plotted in Figure 11.

The accuracy of the decision tree algorithm at different depths is shown in Figure 12. The accuracy of the gradient boost algorithm at different learning rates is shown in Figure 13. The number of estimators was increased from 100 to 500 and the accuracy was measured at different learning rates. It can be observed that the accuracy is the same at numberofestimators=400 for all three learning rates. Later, there is an improvement in the accuracy only for learning rate 0.1 at numberofestimators=500.

A comparison of the performance of the existing approaches that were evaluated on the UPfall dataset is given in Table 3. It can be observed that our proposed methodology has achieved the highest scores in terms of accuracy, precision, sensitivity, and f1-score. The results obtained from the system are shown in Figure 14 and Figure 15. These results represent the sequence of frames of videos that were classified as falling and daily living activities respectively.

## 5. Conclusions and Future Work

This work presents a precise solution for detecting falls and classifying poses by extracting human skeleton features and analyzing their body geometry. Our model which was evaluated on the UPfall dataset achieves an accuracy of 98.32%, which demonstrates its effectiveness in classifying falls and an F1-score of 98.11%. Also, our proposed feature descriptor is invariant to the gender and age of a subject during fall detection. Unlike previous approaches, we do not rely on static techniques such as human silhouette extraction or event-based detection for fall detection, as these methods may not be robust enough to handle changes in the operating environment. Our proposed solution for fall classification has the potential to lower the computational time owing to the features extracted. However, two key constraints of any fall detection system that employs RGB data are preserving the user’s privacy, which can be overcome by incorporating depth cameras, and dealing with low lighting conditions, which can be improved by using night vision cameras.

## Figures and Tables

**Figure 1 sensors-23-06283-f001:**
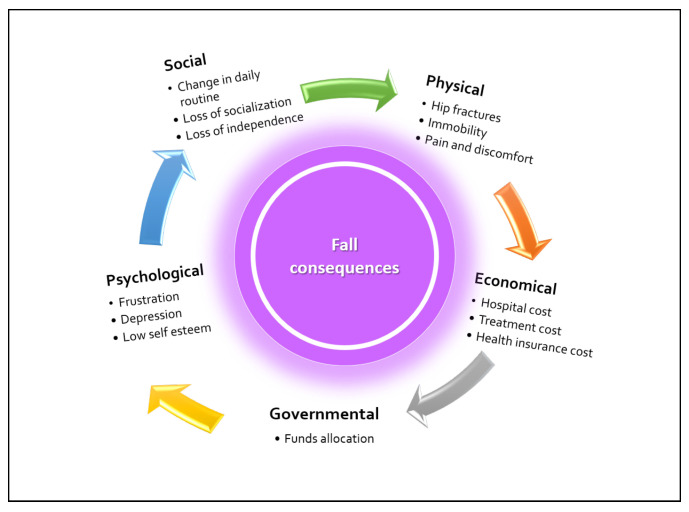
Major consequences of falls.

**Figure 2 sensors-23-06283-f002:**
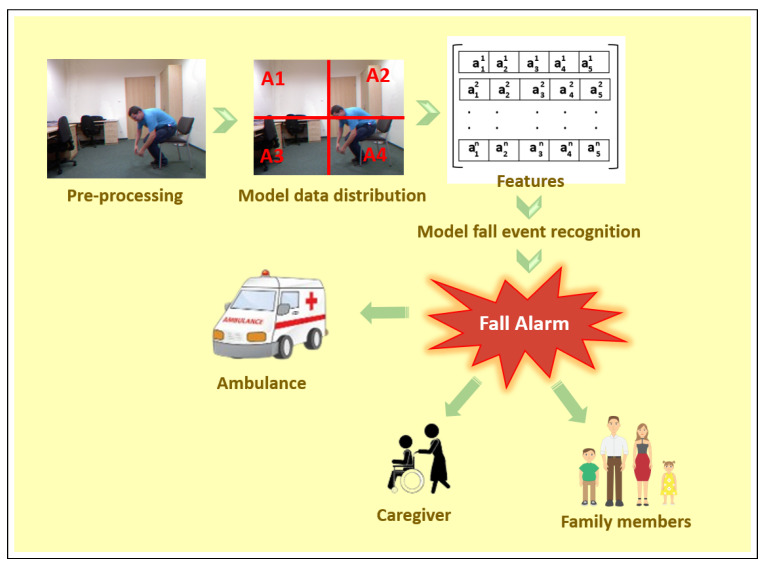
A general fall detection system.

**Figure 3 sensors-23-06283-f003:**
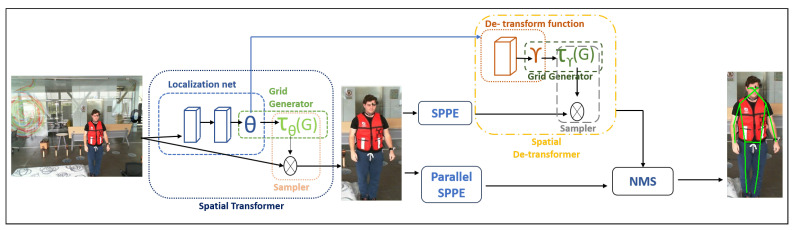
AlphaPose network.

**Figure 4 sensors-23-06283-f004:**
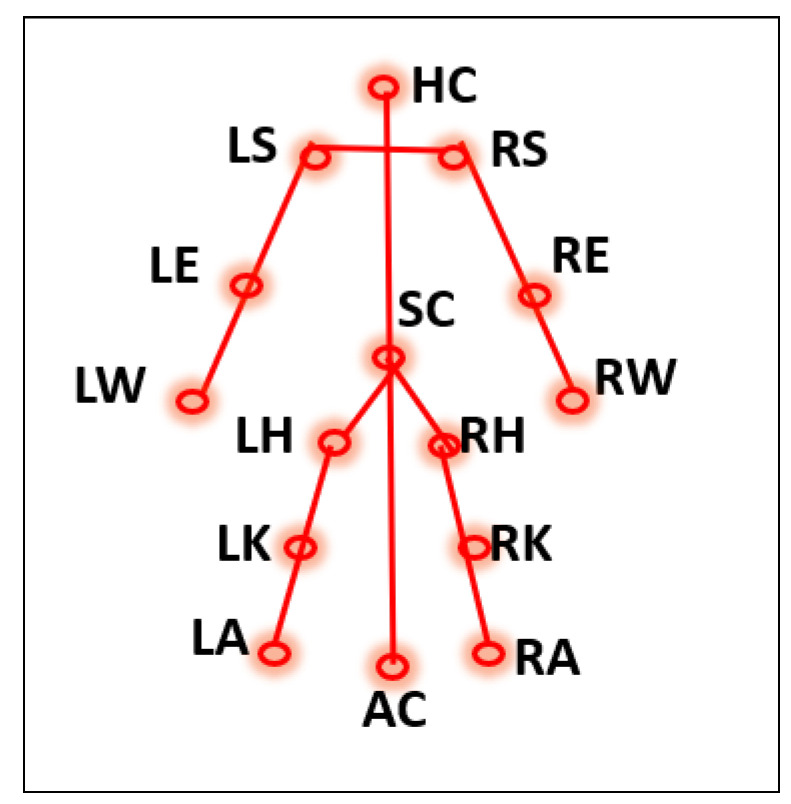
Keypoints considered.

**Figure 5 sensors-23-06283-f005:**
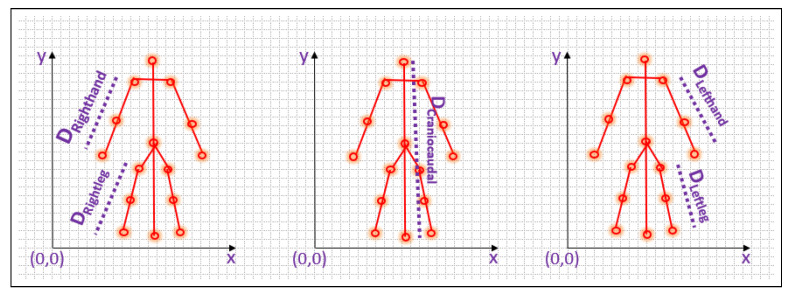
Distance calculation on all segments.

**Figure 6 sensors-23-06283-f006:**
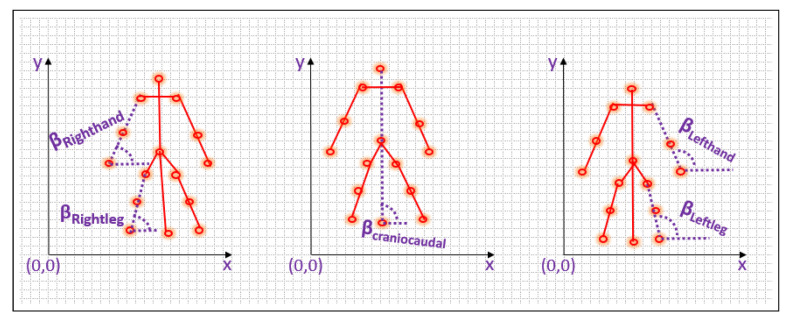
Angle of inclination for all segments.

**Figure 7 sensors-23-06283-f007:**
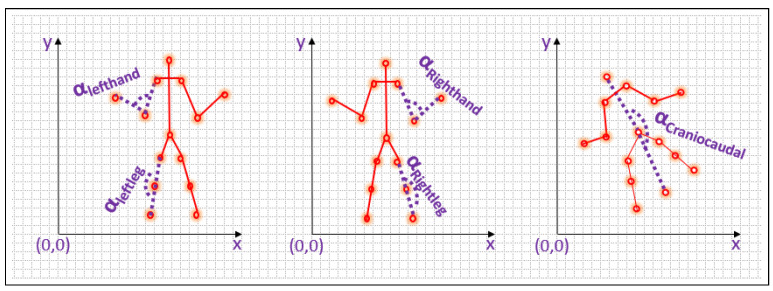
Angle between the non-collinear points of every segment.

**Figure 8 sensors-23-06283-f008:**
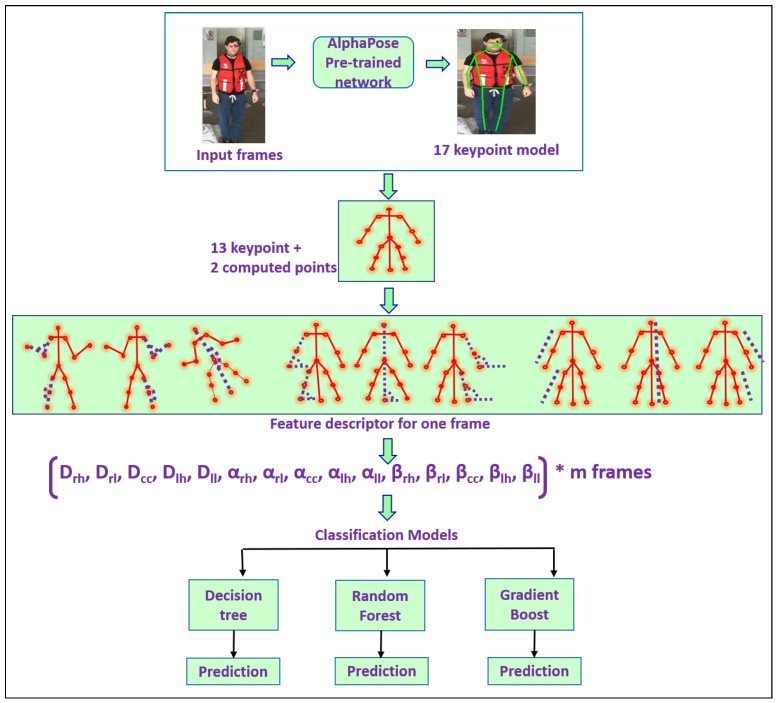
Workflow of the proposed fall detection system.

**Figure 9 sensors-23-06283-f009:**
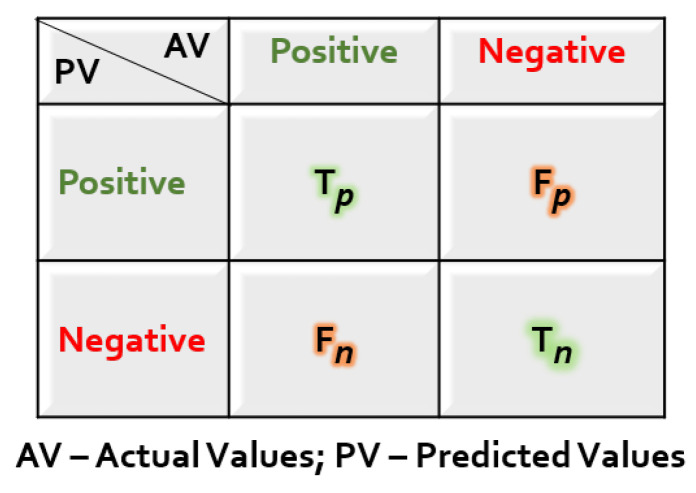
General form of a confusion matrix.

**Figure 10 sensors-23-06283-f010:**
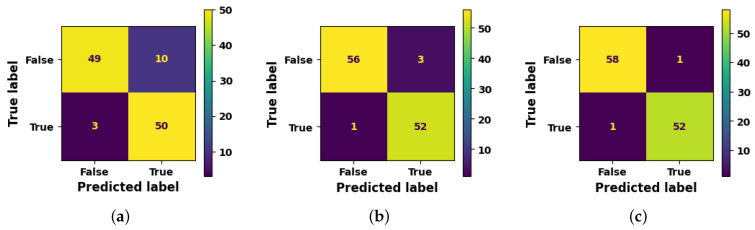
Confusion matrices of the algorithms in the proposed work. (**a**) depicts the confusion matrix of the Decision tree algorithm, (**b**) depicts the confusion matrix of the Random Forest algorithm and (**c**) depicts the confusion matrix of the Gradient boost algorithm.

**Figure 11 sensors-23-06283-f011:**
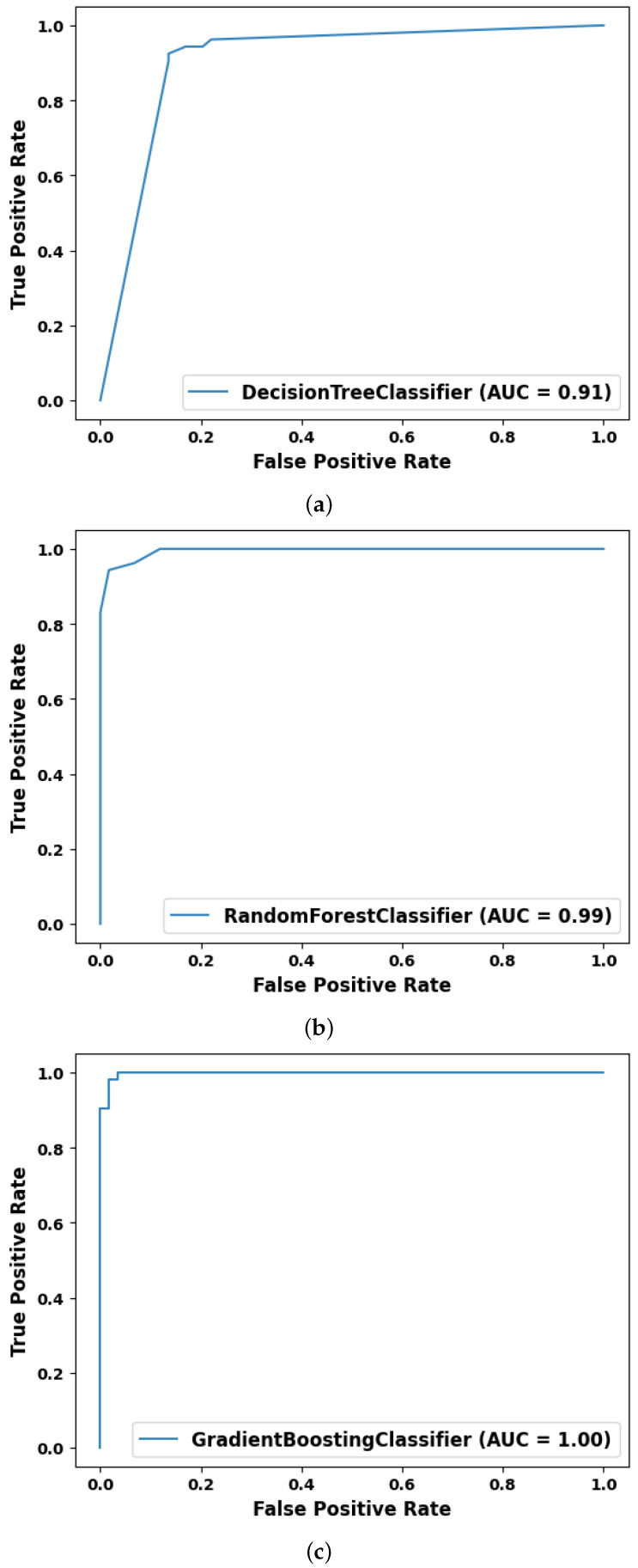
AUC-ROC Curves of the algorithms in the proposed work. (**a**) Decision tree algorithm. (**b**) Random forest algorithm. (**c**) Gradient boost algorithm.

**Figure 12 sensors-23-06283-f012:**
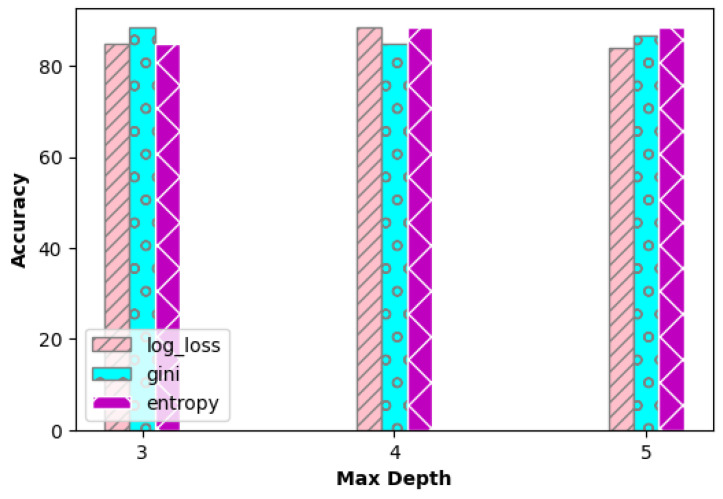
Accuracy of decision tree at different depth.

**Figure 13 sensors-23-06283-f013:**
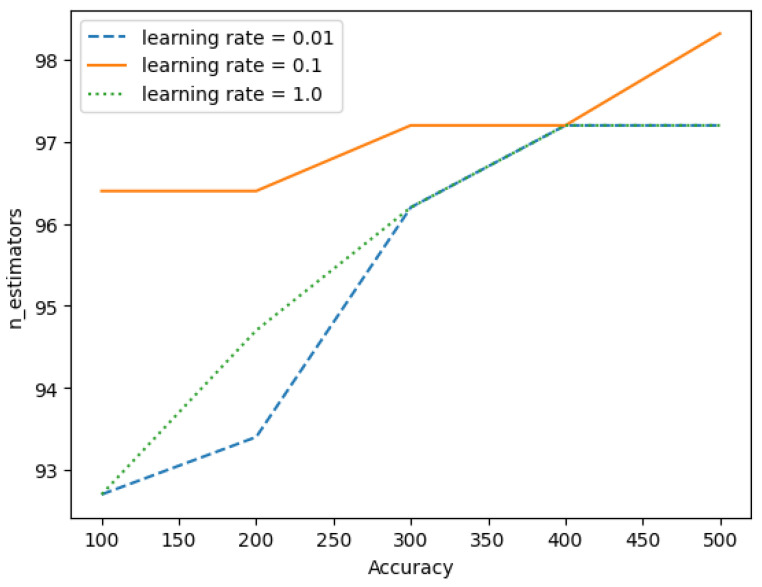
Performance of Gradient Boost algorithm at different learning rates.

**Figure 14 sensors-23-06283-f014:**
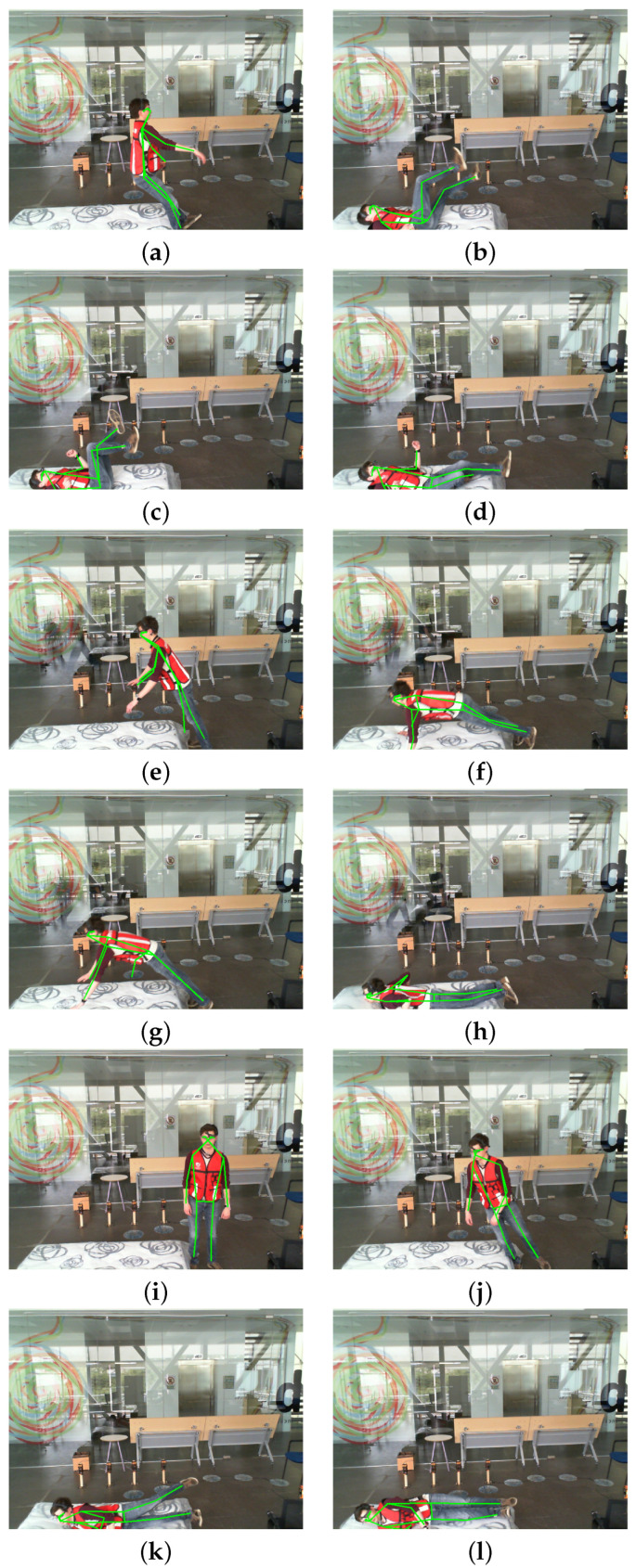
Results of the falling activity of the proposed system. (**a**–**d**) show the sequence of falling backwards, (**e**–**h**) show the sequence of falling forwards and (**i**–**l**) show the sequence of a falling sideways.

**Figure 15 sensors-23-06283-f015:**
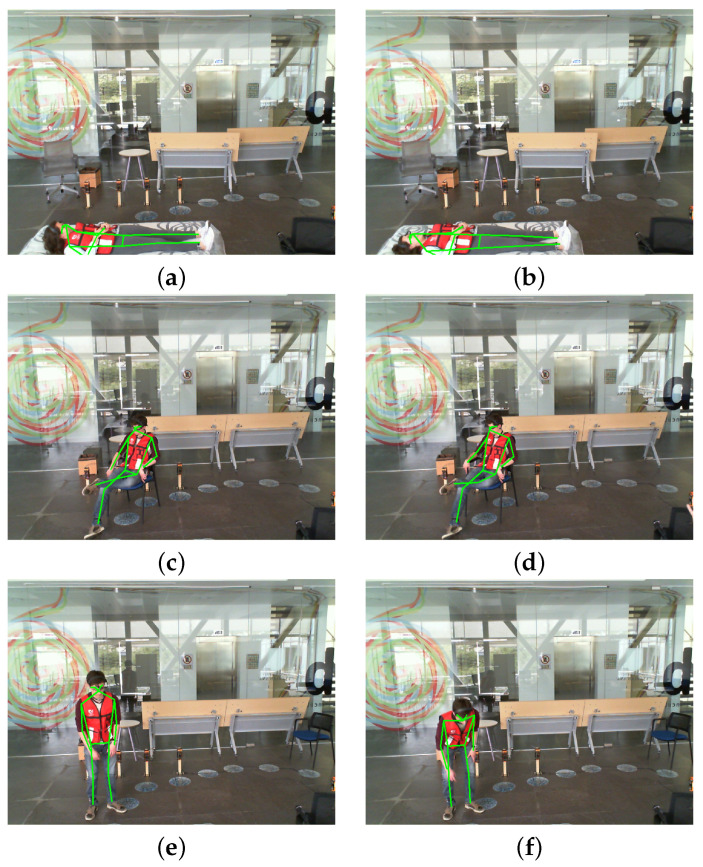
Results of the daily living activities of the proposed system. (**a**,**b**) show images of laying activity, (**c**,**d**) show images of sitting activity and (**e**,**f**) show images of picking up an object.

**Table 1 sensors-23-06283-t001:** An overview of existing vision-based approaches.

Approach	Method	Algorithm
[29]	Head and Torso tracking	Convolutional neural network
[30]	Stacked human silhouette	Binary motion is observed
[31]	Pixel-wise multi scaling skip connection network	Conv LSTM
[32]	Skeleton keypoints using AlphaPose	Random forest, Support vector machine, k-Nearest neighbors, Multi layer perceptron
[33]	Distance and angle between same keypoints in successive frames.	Random forest, Support vector machine, k-Nearest neighbors, Multi layer perceptron

**Table 2 sensors-23-06283-t002:** Performance measurement of all three algorithms.

	Accuracy (%)	Precision (%)	Sensitivity (%)	Specificity (%)	F1-Score (%)
Decision Tree	88.39	84.48	92.45	84.74	88.28
Random Forest	96.43	98.03	94.33	98.30	96.15
Gradient Boost	98.32	98.11	98.11	98.30	98.11

**Table 3 sensors-23-06283-t003:** Comparison of the performance of the existing approaches that were evaluated on UPfall dataset.

Author	Classifier	Sensitivity (%)	Specificity (%)	Precision (%)	F1-Score (%)	Accuracy (%)
[39]	Convolutional neural network (CNN) Lateral camera (cam1)	97.72	81.58	95.24	97.20	95.24
	Convolutional neural network (CNN) Frontal camera (cam2)	95.57	79.67	96.30	96.93	94.78
[36]	K-Nearest Neighbors (KNN)	15.54	93.09	15.32	15.19	34.03
	Support Vector Machine (SVM)	14.30	92.97	13.81	13.83	34.40
	Random Forest (RF)	14.48	92.9	14.45	14.38	32.33
	Multilayer Perceptron (MLP)	10.59	92.21	8.59	7.31	34.03
	Convolutional neural network (CNN)	71.3	**99.5**	71.8	71.2	95.1
[40]	Convolutional neural network (CNN)	97.95	83.08	96.91	97.43	95.64
[32]	Average(RF, SVM, MLP, KNN)	96.80	99.11	96.94	96.87	97.59
[8]	Convolutional neural network (CNN) + Long-short term memory (LSTM)	94.37	98.96	91.08	92.47	96.72
[31]	ConvLSTM	97.68	-	97.71	97.68	97.68
**Our Proposed Work**	Gradient Boost (GB)	**98.11**	98.30	**98.11**	**98.11**	**98.32**

## Data Availability

Not applicable.

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
