# Peer review of "Synergistic Integration of Skeletal Kinematic Features for Vision-Based Fall Detection"

_sensors, 2023, doi:10.3390/s23146283_

Round 1

Reviewer 1 Report

This paper proposes a method to handle the fall detection problem through skeletal kinematic features through cameras. In general, this paper is of limited theoretical contributions. And my primary concerns are as follows.

(1)    Although the authors achieve decent measurements compared with six other methods involving classic machine-learning-based or deep-learning-based classifiers, the contribution to improving the classifier, GB (gradient boost), is limited. Therefore, the improvement is critical, especially from the perspective of pattern recognition.

(2)    The contribution to the pattern should be recognized. However, the authors must clarify that these adopted patterns are first proposed by themselves. At least, they need to clarify which are new and which were proposed by others.

(3)    The contributions in the abstract and the introduction is not clear. Please explain clearly.

(4)    Some parts are confusing. For instance, Lines 57 to 63 on Page 3. Authors claim that recent work is exhibited in Section 2, but they titled this part 1. Related Work.

 Since this work is under the pattern recognition framework, solid contributions to the patterns (features) and the classifier (recognizer) are significant. Otherwise, contributions are not essential enough.

I am satisfied with the language, which does not stand in the way while I am reading. 

Author Response

This paper proposes a method to handle the fall detection problem through skeletal kinematic features through cameras. In general, this paper is of limited theoretical contributions. And my primary concerns are as follows.

Comment 1:    Although the authors achieve decent measurements compared with six other methods involving classic machine-learning-based or deep-learning-based classifiers, the contribution to improving the classifier, GB (gradient boost), is limited. Therefore, the improvement is critical, especially from the perspective of pattern recognition.

Reply 1: The reviewer may kindly note that in the proposed scheme, we defined a set of new informative features by considering the following:

  • Segmenting the human skeleton into five parts(left hand, right hand, left leg, right leg and a craniocaudal segment.)
  • Extracting the distances from the segmented parts.
  • Calculating the angles within the segments.
  • Calculating the angle of inclination for every segment.
  • To capture temporal dynamics, the extracted spatial features are arranged in the temporal sequence. As a result, the feature descriptor in the proposed approach preserves the spatio-temporal dynamics.

The reviewer may kindly note that our main contribution lies in the design of the feature vector and a new fall detection framework. The experimental study shows that our newly derived informative features are more robust in distinguishing falls and fall-like activities.  The manuscript is updated with these contributions. 

Comment 2:  The contribution to the pattern should be recognized. However, the authors must clarify that these adopted patterns are first proposed by themselves. At least, they need to clarify which are new and which were proposed by others.

Reply 2: The reviewer may kindly note that in the literature, 

  • The skeleton key points are used to identify a pattern as fall or not [Reference: 32]
  • Another work in the literature has calculated the distances and angles formed between the same keypoints in successive frames (i.e., temporal domain information) [Reference: 33]
  • It can be noted that both spatial information and temporal information are important in fall detection systems. 

In our work,

  •  We defined a new feature descriptor by using a set of informative features such as calculating the distances between keypoints, angle of inclination and angle between the limbs within a frame (Spatial domain).
  • To capture temporal dynamics, the extracted spatial features from all frames are arranged in their temporal order. As a result, the feature descriptor in the proposed approach preserves the spatio-temporal dynamics.

References: 

[32] Ramirez, H.; Velastin, S.A.; Meza, I.; Fabregas, E.; Makris, D.; Farias, G. Fall detection and activity recognition using human 437 skeleton features. IEEE Access 2021, 9, 33532–33542. 438

[33] Alaoui, A.Y.; El Fkihi, S.; Thami, R.O.H. Fall detection for elderly people using the variation of key points of human skeleton. 439

IEEE Access 2019, 7, 154786–154795.

Comment 3: The contributions in the abstract and the introduction are not clear. Please explain clearly.

Reply 3: In the revised manuscript, we listed the contributions in the abstract and introduction section. We thank the reviewer for the valuable suggestion. 

Comment 4:    Some parts are confusing. For instance, Lines 57 to 63 on Page 3. Authors claim that recent work is exhibited in Section 2, but they titled this part 1. Related Work.

Reply 4: As per the valuable suggestion of the reviewer, necessary modifications have been made to eliminate any inconsistencies.

Comment 5: Since this work is under the pattern recognition framework, solid contributions to the patterns (features) and the classifier (recognizer) are significant. Otherwise, contributions are not essential enough.

Reply 5:

  • The reviewer may kindly note that we defined a new feature descriptor by using a set of informative features that are obtained by segmenting the human skeleton into five parts. These segmented parts represent the hands, legs and a craniocaudal section joining the head to toe. 
  • The novel feature descriptor is computed by calculating the distances within a segment, the angle of inclination for every segment and angles formed by three non collinear points within a segment. 
  • These features are calculated for every frame (Spatial domain).
  • To capture temporal dynamics, the extracted spatial features are arranged in the temporal sequence. As a result, the feature descriptor in the proposed approach preserves the spatio-temporal dynamics.

Reviewer 2 Report

1.  The research aims to automate fall detection with the assistance of cameras. In the proposed method, body geometry is analyzed on the person captured by the camera. The human skeleton is segmented into five parts, each part represented by three non-collinear points (NCP). These points represent the human joints, and their location is derived with the help of the Alpha-pose pose estimation algorithm.

2.     In the figure 3, Alphapose network should be demonstrated in detail.

3.  In the figure 8, workflow of the fall detection system should be demonstrated in detail.

4.    In the figure 17, performance of Gradient Boost algorithm at different learning rates should be demonstrated in detail.

5.   The manuscript has 17 figure s; the number of the figure s should be decreased.

6. The authors are suggested to highlight the contributions of the proposed work, compared to the prior works. A detailed discussion about prior works are suggested to add.

7.      The manuscript has 16 pages; the number of the pages should be increased.

8.      Revise the English thoroughly before submission.

Moderate editing of English language required.

Author Response

The research aims to automate fall detection with the assistance of cameras. In the proposed method, body geometry is analyzed on the person captured by the camera. The human skeleton is segmented into five parts, each part  represented by three non-collinear points (NCP). These points represent the human joints, and their location is derived with the help of the Alpha-pose pose estimation algorithm.

Comment 1: In the figure 3, Alphapose network should be demonstrated in detail.

Reply 1:  As per the reviewer’s suggestion, the diagram (Figure 3) has been modified in the revised manuscript.

Comment 2: In the figure 8, workflow of the fall detection system should be demonstrated in detail.

Reply 2: As per the reviewer’s kind suggestion, the workflow diagram (Figure 8) is updated with more details . 

Comment 3: In the figure 17, performance of Gradient Boost algorithm at different learning rates should be demonstrated in detail.

Reply 3: We have explained in detail about Figure 17 in section 3.4.1.

Comment 4: The manuscript has 17 figures; the number of the figures should be decreased.

Reply 4: As per your valuable suggestion, figures that are a general representation have been removed.

Comment 5: The authors are suggested to highlight the contributions of the proposed work, compared to the prior works. A detailed discussion about prior works are suggested to add.

Reply 5: A list of contributions have been updated in the introduction section. Also, details of other works have been discussed in related work.

Comment 6: The manuscript has 16 pages; the number of the pages should be increased.

Reply 6: We have revised the manuscript and a few additional concepts while addressing the comments from the reviewers. Now, the revised manuscript consists of 19 pages. 

Comment 7: Revise the English thoroughly before submission.

Reply 7: As per the kind reviewer’s suggestion, we have done spelling and grammar corrections to improved the readability by rephrasing some of the statements. Hope the revised manuscript is improved to consider for further consideration.

Author Response

The paper presents the evaluation of a fall detection algorithm using machine learning, based on the Alpha pose algorithm.

Comment 1: Please revise the English throughout the paper as even though the spelling is correct some expressions are poorly used along with the construction of sentences.

Reply 1: As per the reviewer’s suggestion, we have revised the manuscript.

Comment 2: In figure 1, please use another formulation for the Government as the term support is ambiguous and does not express anything in particular.

Reply 2: We have modified the figure and replaced the term “support” with funds allocation.

Comment 3: Please change throughout the paper the word “scheme” when referring to other publications as it is really confusing.

Reply  3: As per your suggestion, the word “Scheme” has been removed.

Comment 4: When discussing the proposed approach please detail a bit more the proposed setup and the environment requirements. Number of cameras, minimum and maximum distance between the camera and the subject.

Reply 4: Thank you for your valuable suggestion. We used the benchmark dataset (UPfall dataset)to evaluate our proposed approach. The video is captured using two cameras (frontal and lateral). The details of the dataset have been updated in the revised manuscript in section 2.1

Comment 5: You are speaking about minimizing the response time. As you are aiming to detect falls and not trying to identify patterns issuing alarms to try and prevent them, what is the general response time and what is your minimization target?

Reply 5: 

  • The response time of fall detection systems that are existing in the market is approximately 30- 45 seconds. (Eg., Philips lifeline, Medical guardian, Bay alarm medical etc.). 
  • In our system, we consider 100 frames and the frame rate is 30fps. So, we are capturing 100 frames in 3.5 seconds. The algorithm runs for approximately 10-12 seconds. Total computation time is 15-16 seconds that nearly halves the existing time.

Comment 6: When you classify the incidents determined by falls (injuries, death) how do you aim to reduce their effects?

Reply 6: Our method not only attempts to categorize, but we also suggest raising an alarm and informing the carers of the incidence. This might be enhanced to alert the closest hospital. This enables prompt action to be taken in the event of a fall, thus reducing its effects.

Comment 7: While wearable devices might have less accuracy, they are not dependent on the environment and thus, can be used also in open spaces. I believe that this should be mentioned.

Reply 7: It can be noted that this point has been mentioned in the revised manuscript.

Comment 8: In figure 3, please be consistent with your notations. AlphaPose or Alphapose, SSTN versus STN

Reply 8: Necessary modifications have been made as suggested by the reviewer.

Comment 9: When defining the selected keypoints from the Alphapose set please illustrate also (in figure 4) the complete dataset and justify the transition from 17 to 13 (+2).

Reply 9: The figure has been modified as per the reviewer’s suggestion. 

  • The major goal in the transition of keypoints is to segment the human skeleton into five sections. 
  • These sections represent two hands, legs and a craniocaudal section that runs from the head to toe. 
  • In fall detection, the craniocaudal section plays a major role. The inclination angle made by this section is crucial for analyzing falls. 

Comment 10: The Spine Center (SC) is located at the middle of the spine, or in the lumbar area? Based on its name it should be located higher than illustrated based on the calculations, it is not the center of the spine. Based on the definition of the Ankle Center, SC should be Hip Center. Also, please add the OXY coordinate system on the figure and motivate the use of two coordinates while specifying the spatial position. Comment also on the situation when the body is rotated with respect to the camera. Define the abbreviations when you use them for the first time.

Reply 10: 

  • The spine center is modified as hip/spine center.
  • The OXY coordinate system has been updated in Figures 5, 6 and 7. 
  • We have used the UPfall dataset that consists of frontal view and lateral view. Also when the body is rotated, some keypoints were missing and these keypoints were calculated using equation 2 and equation 3.

Comment 11: Equations (12) - (16) are “exposed” to the specific limitations of the arctangent. strongly suggest the use of the function which mitigates all the limitations of atan, or alternatively justify the management of all the possible scenarios for the atan function. For the next set atan2 was used..was there a reason for the different selection of the functions?

Reply 11: The reason to use atan are,

  • Values in the second quadrant result in negative numbers and values in the third quadrant result in positive numbers. 
  • This variation from positive to negative and again positive gives a chance to identify some deviation in a regular pattern which is consistently either positive or negative.
  • Identifying this deviation is more crucial for angle of inclination rather than the angle between the limbs. 

Comment 12: When you define the rules for the “fall, non-fall” you should explain them. What is the idea of figure 10? It represents basically the same thing four times.

Reply 12: Figure 10 is a general representation of a random forest algorithm. A random forest is a collection of multiple trees and each tree gives a prediction. The prediction with maximum vote is considered as the prediction of the system. This figure is removed from the paper since it is a general representation. 

Comment 13: Figure 15, 16...there is no page limitation for the paper...please increase the size of the figure to allow the reader to actually see the contents.

Reply 13: The size of figures is increased as per your valuable suggestion.

Comment 14: Figure 17...use different line types and thickness to ensure that all the three lines are visible.

Reply 14: As per the suggestion of the reviewer, the line types have been modified for better readability.

Comment 15: As you target specifically the fall detection for elderly people, please comment on the database content regarding the fall patterns. Or, if there is no difference between age (and gender) specify that also.

Reply 15: Our proposed feature descriptor is invariant to gender and age of a subject while detecting falls.

Round 2

Reviewer 1 Report

According to the improvement and modifications in the revised manuscript, my concerns have been addressed to a large extent. Therefore, I have no more questions, and the paper meets the publication standard of this journal. 

Author Response

Comment 1: According to the improvement and modifications in the revised manuscript, my concerns have been addressed to a large extent. Therefore, I have no more questions, and the paper meets the publication standard of this journal.

Reply 1: We thank the reviewer for their valuable time and suggestions which helped us to improve the manuscript.

Reviewer 2 Report

no further comment.

Author Response

Comment 1: no further comment.

Reply 1: We thank the reviewer for giving detailed comments during the first-round review which helped us to improve the manuscript. We thank the reviewer for their valuable time and constructive feedback throughout the process.

Reviewer 3 Report

The manuscript has undergone significant improvements. The authors have diligently incorporated the feedback provided and have made notable enhancements throughout the document.

I still recommend authors to address comment 3 and remove the word "scheme" when they mention a reference. In a scientific paper, this word means something different ;)

Other comments are related to minor errors which should be resolved (lowercase letter after the dot, lowercase letter at the beginning of the description from Figure 4, Table 1in the middle of the section, and pictures right in the reference section).

I also recommend improving the general appearance of the paper.

In general, the paper showcases an improvement in English usage. However, I would like to suggest that the authors thoroughly review the entire paper and provide a more comprehensive description of their work. This will help enhance the clarity and depth of their findings.

Author Response

The manuscript has undergone significant improvements. The authors have diligently incorporated the feedback provided and have made notable enhancements throughout the document.

Comment 1: I still recommend authors to address comment 3 and remove the word "scheme" when they mention a reference. In a scientific paper, this word means something different ;)

Reply 1: We thank the reviewer for the valuable suggestion. The word “scheme” has been removed from the paper.

Comment 2: Other comments are related to minor errors which should be resolved (lowercase letter after the dot, lowercase letter at the beginning of the description from Figure 4, Table 1 in the middle of the section, and pictures right in the reference section).

Reply 2: We thank the reviewer for these constructive comments and the manuscript is revised accordingly.

Comment 3: I also recommend improving the general appearance of the paper.

Reply 3: The general appearance of the paper has been modified in the revised manuscript based on the valuable comment from the reviewer.

  • The Figures and Tables are arranged in a better way in the revised manuscript to improve the general appearance of the manuscript.
  • “Related Work” has been revised to “ Literature Review”.
  • “Sensor devices” in 1.1 has been updated to “Sensor-based Technology”.
  • “Camera devices” in 1.2 has been updated to “Vision-based Technology”
  • “Experimental analysis” has been modified to “Experimental Study and Result Analysis”.
  • “Conclusion” section has been revised to “Conclusion and Future Work”.

         We thank the reviewer for the valuable suggestion.

Comments on the Quality of English Language

Comment: In general, the paper showcases an improvement in English usage. However, I would like to suggest that the authors thoroughly review the entire paper and provide a more comprehensive description of their work. This will help enhance the clarity and depth of their findings.

Reply: As per the reviewer’s suggestion,  we have done a thorough proofreading and language correction. In addition, a detailed description of the entire procedure has been included in the introduction section of the revised manuscript. We hope currently, the manuscript is in a form to consider for publication.